# A Human Gait Tracking System Using Dual Foot-Mounted IMU and Multiple 2D LiDARs

**DOI:** 10.3390/s22176368

**Published:** 2022-08-24

**Authors:** Huu Toan Duong, Young Soo Suh

**Affiliations:** Department of Electrical, Electronic and Computer Engineering, University of Ulsan, Ulsan 44610, Korea

**Keywords:** human gait tracking, multiple 2D LiDARs, multiple LiDARs calibration, dual foot-mounted IMU

## Abstract

This paper proposes a human gait tracking system using a dual foot-mounted IMU and multiple 2D LiDARs. The combining system aims to overcome the disadvantages of each single sensor system (the short tracking range of the single 2D LiDAR and the drift errors of the IMU system). The LiDARs act as anchors to mitigate the errors of an inertial navigation algorithm. In our system, two 2D LiDARs are used. LiDAR 1 is placed around the starting point, and LiDAR 2 is placed at the ending point (in straight walking) or at the turning point (in rectangular path walking). Using the LiDAR 1, we can estimate the initial headings and positions of each IMU without any calibration process. We also propose a method to calibrate two LiDARs that are placed far apart. Then, the measurement from two LiDARs can be combined in a Kalman filter and the smoother algorithm to correct the two estimated feet trajectories. If straight walking is detected, we update the current stride heading and the foot position using the previous stride headings. Then, it is used as a measurement update in the Kalman filter. In the smoother algorithm, a step width constraint is used as a measurement update. We evaluate the stride length estimation through a straight walking experiment along a corridor. The root mean square errors compared with an optical tracking system are less than 3 cm. The performance of proposed method is also verified with a rectangular path walking experiment.

## 1. Introduction

Human gait monitoring and analysis are important in clinical rehabilitation [1,2], functional mobility diagnostics [3,4], and sports training [5,6,7]. With some technological tools, the gait phase and human gait parameters can be determined through gait analysis. The Vicon optical tracking system [8] and the GAITRite system [9] are the gold standards of gait analysis since they provide accurate measurements. The optical system captures the position of passive reflective markers attached on human body using multi-cameras, while the GAITRite system measures gait parameters via an electronic force platform. However, these standard systems are bulky, expensive, and require a long time for set up and post-processing the data. It may be difficult to deploy these evaluation systems in some rural areas or remote locations.

Wearable sensors, such as inertial sensors or inertial measurement units (IMUs), are used as low-cost alternative instruments in gait analysis [10]. They are small and lightweight, have low power consumption, and are integrated in most smart wearable devices. They can be attached to various parts of the human body including the foot, waist, chest, and head. The gait parameters are then estimated using the recorded movement signal. For example, the walking step length can be estimated with a waist-mounted IMU using a linear regression model [11]. This method requires a calibration process to construct the step length model. A common method for foot-mounted IMU is the numerical integration of inertial data to estimate foot velocity and position. However, the drawback of low-cost IMU is the sensor noises, which result in the accumulated integration errors. A most widely used technique to reduce these errors is the zero-velocity update (ZVU) [12,13], which assumes that the foot velocity is zero when it contacts the ground. Once the stance phase is detected, a ZVU is used as a pseudo measurement of the Kalman filter to reset the errors. However, as the position and heading are still unobservable, the heading drift grows over time. Thus, the positioning accuracy is decreased.

Some researchers use a dual foot-mounted IMU system to reduce the symmetrical heading errors [14,15]. There are two problems in the dual foot-mounted IMU system. The first one is the initial heading and position of each IMU, which is usually solved using a calibration process. The other one is the heading drift over time. Using the mapping information, a heuristic drift elimination (HDE) algorithm is proposed to update the heading with the dominant direction of building [16,17]. This method cannot apply to large space, where the building direction is not available. Based on the relative position of two feet, some distance constraints are proposed as the measurements in the Kalman filter frame work [18,19,20,21,22]. These methods update the stride heading and foot position if straight walking is detected or use the maximum range of stride length or two feet step length to constrain the divergence of the moving foot. However, the heading cannot be effectively corrected, especially in the turning phase.

In human gait analysis, a 2D LiDAR is also used to track the lower limb. The LiDAR does not require external markers and has simple installation. It can be mounted on a moving platform such as a mobile robot [23,24] or a smart walker [25,26], or can be fixed on the ground at a height of human shin [27,28,29,30]. Using the LiDAR with moving platform requires more additional sensors as well as a control system. Thus, the system is complicated and bulky. The system with the LiDAR is installed on the ground is simple. It scans human leg and estimates the leg center point during walking. The LiDAR-based human leg estimation is quite accurate and contains the walking direction with respect to the local environment. However, the effective human leg tracking range of a single 2D LiDAR is limited within a few meters. In [31], a fusion system of multiple 2D LiDARs has been proposed to extend the human leg tracking range. However, the LiDARs cannot be placed far apart to guarantee the continuity of leg scan data. Therefore, the tracking range is still limited. For example, for 20 m of straight walking, three LiDARs need to be used.

The aim of this paper is to combine multiple 2D LiDARs with a dual foot-mounted IMU system to overcome the disadvantage of each single system. The proposed system can increase the human gait tracking range and reduce the heading drift of the inertial sensor system. Two 2D LiDARs are used in this paper. The first one is placed around the starting point. The other one is placed based on the walking route: at the ending point in straight walking or at the turning point in rectangular path walking. The LiDAR-based human leg center point estimation is then used as a measurement update of the inertial sensor system. The main contributions of this paper are as follows:Multiple 2D LiDARs are combined in the dual foot-mounted IMU system for heading and position correction;The initial heading and position of each IMU are automatically estimated using the first LiDAR without any calibration process;A calibration algorithm of two 2D LiDARs (placed far apart) is used, and the calibration parameters are then used to transform the human leg center point into the same coordinate system;In contrast with the HDE algorithm, which uses the building heading information, we propose a simple walking stride heading update algorithm when straight walking is detected. The foot position estimation from the updated heading is then used as a measurement update of the Kalman filter;The estimated errors of the Kalman filter are then compensated using a quadratic optimization-based smoother algorithm [32]. A constraint of the step width is also proposed in the smoother algorithm if the straight walking is detected.

## 2. System Overview

The proposed system is configured, as shown in Figure 1. Two foot-mounted IMU systems are synchronized and consist of a microSD card to store the initial data. Two 2D LiDARs are horizontally placed on the ground to scan human leg. The data from IMUs and LiDARs are then post-processed using Matlab.

As we can see in Figure 1, there are four sensor coordinate systems: two IMUs and two LiDARs. To track the human gait, we define the global coordinate system and two foot body coordinate systems. The left/right foot body coordinate systems are set to coincide with corresponding left/right IMU coordinate system. Subscripts *L* and *R* denote the left and right side, respectively. The three axes of the global coordinate system are chosen as follows: *x* and *y* axes coincide with the LiDAR 1 coordinate system, the *z* axis points upwards, and the origin is the projection of LiDAR 1 origin to the horizontal plane of two IMUs. The height of the global coordinate system is shifted so that the initial z−positions of two feet are zero. For a given vector, we sometimes use subscript *b* (body) and *w* (global) to emphasize that a vector is expressed in the particular coordinate.

The inertial sensor outputs are acceleration $y_{a}\in R^{3}$ and angular velocity $y_{g}\in R^{3}$, and are modeled as follows:(1)$$\begin{array}{cc} y_{a} & =a_{b}+C_{w}^{b}\left(t\right)\tilde{g}+v_{a}\left(t\right),\\ y_{g} & =\omega_{b}\left(t\right)+v_{g}\left(t\right), \end{array}$$
where $a_{b}\in R^{3}$ is the acceleration produced by forces other than the gravitational field; $\tilde{g}={\left[\begin{array}{ccc} 0 & 0 & 9.8 \end{array}\right]}^{T}$ is the gravitational vector in the global coordinate system; $C_{w}^{b}\in R^{3\times 3}$ is the rotation matrix from global coordinate system to the body coordinate system; $\omega_{b}\in R^{3}$ is the body angular velocity; and $v_{a}\in R^{3}$ and $v_{g}\in R^{3}$ are the accelerometer and gyroscope measurement noises, respectively. We assume that $v_{a}$ and $v_{g}$ are white Gaussian noises, where their covariances are given by:(2)$$R_{a}=E\left\{v_{a}v_{a}^{T}\right\},\;R_{g}=E\left\{v_{g}v_{g}^{T}\right\}.$$

The block diagram of our proposed system is shown in Figure 2. Firstly, the data from sensors is preprocessed. The single LiDAR processing module estimates the human leg trajectories using LiDAR scan data. The leg positions are expressed in each LiDAR coordinate system. The first calibration algorithm calibrates two LiDARs coordinate systems to transform the leg positions from LiDAR 2 into the LiDAR 1 coordinate system. The second calibration algorithm uses the IMUs and LiDAR 1 data to output the initial rotation angles and position of each IMU in the LiDAR 1 coordinate system. Then, the Kalman filter and the smoother algorithm are used to estimate the two feet trajectories. In the Kalman filter, the zero-velocity update (ZVU) and the LiDAR-based stance foot positions are used as a measurement update. A simple straight walking detection algorithm is also proposed to correct the heading estimation and the foot position. A smoother algorithm is then used to reduce the estimation error and smooth the trajectories of the two feet. In this smoother algorithm, the straight-walking-based heading and position update is replaced by the step width constraint measurement update.

## 3. Data Preprocessing

The aim of this part is to estimate the human leg center point from LiDAR and estimate the initial heading and position of each IMU in the global system. In this paper, the symbols |.| and ||.|| represent the absolute value and the Euclidean norm, respectively. We also denote the estimated value, the true value, and the error of *x* as $\hat{x}$, $\tilde{x}$, and $x_{e}$, respectively, where $\tilde{x}=\hat{x}+x_{e}$.

### 3.1. Single LiDAR-Based Human Leg Center Point Estimation

The LiDAR scan data are then used to estimate the human leg center point trajectories using the algorithm in [27,31]. The outputs of this module are the human leg positions in stance phases which are represented by diamond shapes in Figure 3.

Let $p_{i,j,k_{j}}^{L_{i}}$ denote the positions of left and right legs positions in the stance phase expressed in the LiDAR *i* coordinate system, where $L_{i}$ represents LiDAR *i*, j=L,R is the left or right leg, and $k_{j}=0,1,2,...$ denotes the sequence of stance phases of *j* leg. For example, $k_{L}=0$ means the left foot is in initial standing still period. Since these leg positions are expressed in different LiDAR coordinate systems, the pose between two LiDARs is computed in the next section.

### 3.2. Calibration 1: Two LiDARs Calibration

The calibration parameters of two LiDARs are the rotation angle and the translation vector. In our previous research of human gait tracking using multiple LiDARs, the working space is shared by the LiDARs; therefore, a cylinder is placed arbitrarily in the common scanning space to calibrate the system. In this research, two LiDARs are placed far apart and there is no common scan data. Therefore, in stead of arbitrary choice, we place the cylinder at some fixed points which are already known their geometric relationship.

The calibration configuration is shown in Figure 4. The cylinder with measured radius ${\hat{r}}_{c}$ is placed at four points $A_{1}$, $B_{1}$ (in LiDAR 1 scanning range), $A_{2}$, and $B_{2}$ (in LiDAR 2 scanning range). These four points are manually calibrated to identify the following parameters: the intersection *O* of $A_{1}B_{1}$ and $A_{2}B_{2}$, the angle $\hat{\alpha }$ between $A_{1}B_{1}$ and $A_{2}B_{2}$, and the corresponding distance from *O* to each point ${\hat{d}}_{m_{i}}$, where m=A,B and i=1,2. We also denote the measurement errors of ${\tilde{r}}_{c}$, $\tilde{\alpha }$, and ${\tilde{d}}_{m_{i}}$ by $r_{c,e}$, $\alpha_{e}$, and $d_{e}$, respectively.

The coordinate systems of two LiDARs are represented by $O_{1}-x_{1}y_{1}$ and $O_{2}-x_{2}y_{2}$. The calibration parameters are the translation vector $T_{12}$ from $O_{1}$ to $O_{2}$ and the rotation angle $\theta_{12}$ from the LiDAR 2 coordinate system to the LiDAR 1 coordinate system. To account the uncertainties of the LiDAR data measurement and the measured configuration parameters, we propose an iterative algorithm to compensate the calibration parameters errors. The state vector $x_{c}$ consists of the calibration parameters and the cylinder’s estimated center points at points $A_{1}$ and $B_{1}$:(3)$$x_{c}={\left[\begin{array}{cccc} \theta_{12} & T_{12}^{T} & {\left(c_{A_{1}}^{L_{1}}\right)}^{T} & {\left(c_{B_{1}}^{L_{1}}\right)}^{T} \end{array}\right]}^{T}\in R^{7}.$$

Let $s_{m_{i},k}^{L_{i}}$ represent the cylinder sample scan data of LiDAR *i* at position $m_{i}$. At each position, the total number of scan data point is $N_{m_{i}}$.

#### 3.2.1. Initialization

We firstly initialize the cylinder’s center points using the least square fitting algorithm. The initial cylinder’s center points ${\hat{c}}_{m_{i},0}^{L_{i}}$ are then used to estimate the initial calibration parameters as follows:The unit vector is estimated by: ${\hat{v}}^{L_{i}}=\frac{{\hat{c}}_{A_{i},0}^{L_{i}}-{\hat{c}}_{B_{i},0}^{L_{i}}}{∥{\hat{c}}_{A_{i},0}^{L_{i}}-{\hat{c}}_{B_{i},0}^{L_{i}}∥},\;i=1,2$;The initial rotation angle is estimated by: ${\hat{\theta }}_{12,0}=∠\left({\hat{v}}^{L_{1}},-R\left(\hat{\alpha }\right){\hat{v}}^{L_{2}}\right)$;The initial translation vector is estimated by:${\hat{T}}_{12,0}={}^{1}{\hat{c}}_{A_{1},0}+{\hat{d}}_{A_{1}}{\hat{v}}^{L_{1}}+{\hat{d}}_{A_{2}}R\left(\hat{\alpha }\right){\hat{v}}^{L_{1}}-R\left({\hat{\theta }}_{12,0}\right){}^{2}{\hat{c}}_{A_{2},0}$.

In the above equations, R(φ) is the rotation matrix of an angle φ with respect to the *z*-axis: $R\left(\varphi \right)=\left[\begin{array}{cc} \cos\varphi & -\sin\varphi \\ \sin\varphi & \cos\varphi \end{array}\right]$.

#### 3.2.2. Iterative Algorithm

The errors state vector is defined as follows:$$x_{c,e}={\left[\begin{array}{cccc} \theta_{12,e} & T_{12,e}^{T} & {\left(c_{A_{1},e}^{L_{1}}\right)}^{T} & {\left(c_{B_{1},e}^{L_{1}}\right)}^{T} \end{array}\right]}^{T}\in R^{7}.$$

The measurement equations are then identified using the LiDAR scan data.

For LiDAR 1, the circle fitting equation is as follows:(4)$${\tilde{r}}_{c}=∥{\tilde{s}}_{m_{1},k}^{L_{1}}-{\tilde{c}}_{m_{1}}^{L_{1}}∥,\;m=A,B.$$

By inserting the measurement errors, we can identify the measurement equation for each scan data point of LiDAR 1 as follows:(5)$$z_{m_{1},k}={\hat{r}}_{c}-∥s_{m_{1},k}^{L_{1}}-{\hat{c}}_{m_{1}}^{L_{1}}∥=h_{m_{1},k}c_{m_{1},e}^{L_{1}}+\left[\begin{array}{cc} h_{m_{1},k} & -1 \end{array}\right]\left[\begin{array}{c} v_{s_{m_{1},k}}\\ r_{c,e} \end{array}\right],$$
with $h_{m_{1},k}=\frac{-{\left(s_{m_{1},k}^{L_{1}}-{\hat{c}}_{m_{1}}^{L_{1}}\right)}^{T}}{∥s_{m_{1},k}^{L_{1}}-{\hat{c}}_{m_{1}}^{L_{1}}∥}$ and $v_{s_{m_{1},k}}$ is the measurement errors of sample data $s_{m_{1},k}^{L_{1}}$.

For LiDAR 2, we fit the circle after transforming the scan data into the LiDAR 1 coordinate system:(6)$${\tilde{r}}_{c}=∥R\left({\tilde{\theta }}_{12}\right){\tilde{s}}_{m_{2},k}^{L_{2}}+{\tilde{T}}_{12}-{\tilde{c}}_{m_{2}}^{L_{1}}∥,\;m=A,B,$$
where ${\tilde{c}}_{m_{2}}^{L_{1}}={\tilde{c}}_{A_{1}}^{L_{1}}+{\tilde{d}}_{A_{1}}{\tilde{v}}^{L_{1}}+{\tilde{d}}_{m_{2}}R\left(\tilde{\alpha }\right){\tilde{v}}^{L_{1}}$. The small error of unit vector ${\tilde{v}}^{L_{1}}$ can be represented by a small angle $\beta_{e}$ as follows: ${\tilde{v}}^{L_{1}}=R\left(\beta_{e}\right){\hat{v}}^{L_{1}}\approx {\hat{v}}^{L_{1}}+{\hat{u}}^{L_{1}}\beta_{e}$, with ${\hat{u}}^{L_{1}}=\left[\begin{array}{cc} 0 & -1\\ 1 & 0 \end{array}\right]{\hat{v}}^{L_{1}}$. The rotation matrix is approximated as follows: $R\left(\tilde{\theta }\right)=R\left(\hat{\theta }+\theta_{e}\right)\approx R\left(\hat{\theta }\right)+G\left(\hat{\theta }\right)\theta_{e}$, where $G\left(\hat{\theta }\right)=R\left(\hat{\theta }\right)\left[\begin{array}{cc} 0 & -1\\ 1 & 0 \end{array}\right]$. By inserting the measurement errors, (Equation 6) becomes:$$\begin{array}{cc} {\hat{r}}_{c} & +r_{c,e}=∥\left(R\left({\hat{\theta }}_{12}\right)+G\left({\hat{\theta }}_{12}\right)\theta_{e}\right)\left(s_{m_{2},k}^{L_{2}}-v_{s_{m_{2},k}}\right)+{\hat{T}}_{12}+T_{12,e}-({\hat{c}}_{A_{1}}^{L_{1}}+c_{A_{1},e}^{L_{1}}+\\ \; & \left({\hat{d}}_{A_{1}}+d_{e}\right)\left({\hat{v}}^{L_{1}}+{\hat{u}}^{L_{1}}\beta_{e}\right)+\left({\hat{d}}_{m_{2}}+d_{e}\right)\left(R\left(\hat{\alpha }\right)+G\left(\hat{\alpha }\right)\alpha_{e}\right)\left({\hat{v}}^{L_{1}}+{\hat{u}}^{L_{1}}\beta_{e}\right))∥. \end{array}$$

The measurement equation at point $m_{2},m=A,B$ is as follows:(7)$$\begin{array}{cc} \; & z_{m_{2},k}={\hat{r}}_{c}-∥f_{m_{2},k}∥=h_{m_{2},k}(G\left({\hat{\theta }}_{12}\right)s_{m_{2},k}^{L_{2}}\theta_{12,e}+T_{12,e}-c_{A_{1},e}^{L_{1}}-R\left({\hat{\theta }}_{12}\right)v_{s_{m_{2},k}}\\ \; & -\left({\hat{v}}^{L_{1}}+R\left(\hat{\alpha }\right){\hat{v}}^{L_{1}}\right)d_{e}-\left({\hat{d}}_{A_{1}}{\hat{u}}^{L_{1}}+{\hat{d}}_{m_{2}}R\left(\hat{\alpha }\right){\hat{u}}^{L_{1}}\right)\beta_{e}-{\hat{d}}_{m_{2}}G\left(\hat{\alpha }\right){\hat{v}}^{L_{1}}\alpha_{e})-r_{c,e}, \end{array}$$
where $f_{m_{2},k}=R\left({\hat{\theta }}_{12}\right)s_{m_{2},k}^{L_{2}}+{\hat{T}}_{12}-\left({\hat{c}}_{A_{1}}^{L_{1}}+{\hat{d}}_{A_{1}}{\hat{v}}^{L_{1}}+{\hat{d}}_{m_{2}}R\left(\hat{\alpha }\right){\hat{v}}^{L_{1}}\right)$, and

$h_{m_{2},k}=\frac{f_{m_{2},k}^{T}}{∥f_{m_{2},k}∥}$.

By combining (Equation 5) and (Equation 7), the measurement equation for all four positions of cylinder is as follows:(8)$$z_{c}=H_{c}x_{c,e}+B_{c}w_{c},$$
where $z_{c}=\left[\begin{array}{c} {\hat{r}}_{c}-∥s_{m_{1},k}^{L_{1}}-{\hat{c}}_{m_{1}}^{L_{1}}∥\\ {\hat{r}}_{c}-∥f_{m_{2},k}∥ \end{array}\right]\in R^{N_{a}\times 1}$, $w_{c}=\left[\begin{array}{c} v_{s_{m_{1},k}}\\ v_{s_{m_{2},k}}\\ r_{c,e}\\ d_{e}\\ \beta_{e}\\ \alpha_{e} \end{array}\right]\in R^{2N_{a}+4}$, and $H_{c}\in R^{N_{a}\times 7}$ and $B_{c}\in R^{N_{a}\times 2N_{a}+4}$ can be identified using (Equation 5) and (Equation 7). Here, $N_{a}=\sum_{m=(A,B),i=(1,2)}^{}N_{m_{i}}$ is the total number of sample scan data at four positions of cylinder. We assume that the measurement noises are zero mean white Gaussian noises, where covariances are given by:(9)$$\begin{array}{cc} R_{r_{c}} & =E\left\{r_{c,e}^{T}r_{c,e}\right\}\in R,\;R_{d}=E\left\{d_{e}^{T}d_{e}\right\}\in R,\\ R_{\beta } & =E\left\{\beta_{e}^{T}\beta_{e}\right\}\in R,\;R_{\alpha }=E\left\{\alpha_{e}^{T}\alpha_{e}\right\}\in R. \end{array}$$

The covariance of LiDAR measurements can be found in [31]. Then, we can compute the covariance matrix of measurement noises of (Equation 8) as follows:(10)$$Q_{c}=B_{c}E\left\{w_{c}w_{c}^{T}\right\}B_{c}^{T}.$$

The error-state vector is estimated as follows:(11)$$x_{c,e}={\left(H_{c}^{T}Q_{c}^{-1}H_{c}\right)}^{-1}H_{c}^{T}Q_{c}^{-1}z_{c}.$$

The state vector (Equation 3) is then updated using $x_{c,e}$. Using the updated cylinder center points at $A_{1}$ and $B_{1}$, the unit vector ${\hat{v}}^{L_{1}}$ is calculated at each iteration. This process is iterated until a stop condition is satisfied:(12)$$∥x_{c,e}∥<\gamma .$$

Using the calibration parameters, we can transform the stance leg positions from LiDAR 2 coordinate into LiDAR coordinate as follows:(13)$$p_{2,j,k_{j}}^{L_{1}}=\left[\begin{array}{cc} R({\hat{\theta }}_{12}) & {\hat{T}}_{12} \end{array}\right]p_{2,j,k_{j}}^{L_{2}},\;j=L,R.$$

Figure 5 shows an example of transformation result of 20 m straight walking. The stance leg positions are transformed from LiDAR 2 into the LiDAR 1 coordinate system.

### 3.3. Calibration 2: Initial Heading and Position of Each IMU Estimation

The initial rotation angle and position of each IMU in the LiDAR 1 coordinate system are automatically estimated using the first walking stride data from IMUs and LiDAR 1. To estimate the initial rotation angle, we use the TRIAD algorithm [33]. This algorithm requires two pairs of reference and corresponding observation unit vectors. Two reference vectors in the LiDAR 1 coordinate system are gravitational vector $\tilde{g}$ and the first walking stride direction ${\left[\begin{array}{cc} {\left(S_{1,j,1}^{L_{1}}\right)}^{T} & 0 \end{array}\right]}^{T}$ which is estimated from LiDAR 1, where $S_{1,j,1}^{L_{1}}=p_{1,j,1}-p_{1,j,0},\;j=L,R$. Two corresponding observation vectors are the average accelerometer output of one second standing still ${\bar{y}}_{a,j,0}$ and the first walking stride direction estimated from each IMU ${\left[\begin{array}{cc} {\left(S_{j,1}^{I_{j}}\right)}^{T} & 0 \end{array}\right]}^{T}$. To estimate the IMU-based first walking stride direction, a simple inertial navigation algorithm with ZVU is applied to the first walking stride data of each IMU. The initial rotation matrix can be estimated using the following equations:(14)$$\begin{array}{cc} {\bar{y}}_{a,j,0} & =C_{w,0}^{b}\tilde{g},\\ \left[\begin{array}{c} S_{j,1}^{I_{j}}\\ 0 \end{array}\right] & =C_{w,0}^{b}\left[\begin{array}{c} S_{1,j,1}^{L_{1}}\\ 0 \end{array}\right],\;j=L,R. \end{array}$$

After that, we can estimate the initial rotation angles using $C_{w,0}^{b}$.

The initial positions of each IMU ($r_{j,0}={\left[\begin{array}{ccc} x_{j,0} & y_{j,0} & 0 \end{array}\right]}^{T},j=L,R$) are estimated using the estimation of LiDAR-based human leg center points (Figure 6). We assume that the human leg is vertically straight during the initial standing still phase, as well as the mid-stance phase of walking. Therefore, the ankle joint $K_{j}$ can be approximated as the projection of leg center point on the x−y plane. By assuming that the coordinate of $K_{j}$ in IMU *j*, $r_{K_{j}}^{I_{j}}$, is pre-calibrated, the initial position of each IMU in LiDAR 1 coordinate can be computed by:(15)$$r_{j,0}=\left[\begin{array}{c} p_{1,j,0}\\ 0 \end{array}\right]+{\left(C_{w,0}^{b}\right)}^{T}r_{K_{j}}^{I_{j}},\;j=L,R.$$

## 4. Proposed Algorithm

The proposed human gait tracking algorithm can be referred to as an inertial navigation system (INS).

### 4.1. Basic Inertial Navigation System Mechanization

The basic equations [34] for inertial navigation are given as follows:(16)$$\begin{array}{cc} \dot{q} & =\frac{1}{2}\Omega \left(\omega_{b}\right)q,\\ \dot{v} & =C^{T}\left(q\right)a_{b},\\ \dot{r} & =v, \end{array}$$
where $q\in R^{4}$ is the quaternion which represents the attitude of foot, and $v\in R^{3}$ and $r\in R^{3}$ are the velocity and position of foot in global coordinate, respectively. The symbol $\Omega (\omega_{b})$ is defined by:$$\Omega \left(\omega_{b}\right)≜\left[\begin{array}{cccc} 0 & -\omega_{x} & -\omega_{y} & -\omega_{z}\\ \omega_{x} & 0 & \omega_{z} & -\omega_{y}\\ \omega_{y} & -\omega_{z} & 0 & \omega_{x}\\ \omega_{z} & \omega_{y} & -\omega_{x} & 0 \end{array}\right].$$

The inertial navigation algorithm is used to estimate $\hat{q}$, $\hat{v}$ and $\hat{r}$ by integrating (Equation 16). Due to sensor noises in the low-cost inertial sensor, the errors increase quickly if there is no reference measurement. To overcome this problem, a ZVU method is used to reset the accumulated errors. The method assumes that the velocity of the foot is zero when it touches the ground. Thus, once the stance phase is detected, the ZVU can be fused with the INS algorithm by an error-state Kalman filter.

To detect the stance phase during walking, a simple zero-velocity detection algorithm is used [35,36]. A discrete time *k* is determined to belong to the zero-velocity interval if the following conditions are satisfied:(17)$$\begin{array}{cc} ∥y_{g,i}∥ & \leq B_{g},\;k-\frac{N_{g}}{2}\leq i\leq k+\frac{N_{g}}{2}\\ ∥y_{a,i}-y_{a,i-1}∥ & \leq B_{a},\;k-\frac{N_{a}}{2}\leq i\leq k+\frac{N_{a}}{2}, \end{array}$$
where $B_{g}$ and $B_{a}$ are threshold values, and $N_{g}$ and $N_{a}$ are even number integers.

The errors in the numerically integrated values of (Equation 16) are estimated using a Kalman filter. Let $q_{e}\in R^{4}$, $v_{e}\in R^{3}$, and $r_{e}\in R^{3}$ be the estimation errors in $\hat{q}$, $\hat{v}$, and $\hat{r}$, which are defined as follows:(18)$$\begin{array}{cc} q_{e} & ={\hat{q}}^{*}⊗q,\\ v_{e} & =v-\hat{v},\\ r_{e} & =r-\hat{r}, \end{array}$$
where ⊗ is the quaternion multiplication and $q^{*}$ denotes the complex quaternion conjugate of *q*. It is assumed that the quaternion errors are small. Thus, $q_{e}$ can be approximated by:$$q_{e}=\left[\begin{array}{c} 1\\ {\bar{q}}_{e} \end{array}\right]\in \left[\begin{array}{c} R\\ R^{3} \end{array}\right].$$

With this assumption, the attitude error is represented by the three-dimensional vector ${\bar{q}}_{e}$. We define the error-state vector as follows:(19)$$x_{e}=\left[\begin{array}{c} {\bar{q}}_{e}\\ r_{e}\\ v_{e} \end{array}\right]\in R^{9}.$$

The state transition equation is given by:(20)$${\dot{x}}_{e}=Ax+w,$$
where $A=\left[\begin{array}{ccc} \left[-y_{g}\times \right] & 0_{3} & 0_{3}\\ 0_{3} & 0_{3} & I_{3}\\ -2C^{T}\left(\hat{q}\right)\left[y_{a}\times \right] & 0_{3} & 0_{3} \end{array}\right]$ and $w=\left[\begin{array}{c} -0.5v_{g}\\ 0\\ -C^{T}\left(\hat{q}\right)v_{a} \end{array}\right]$. For a vector $p\in R^{3}$, [p×] is defined by:$$\left[p\times \right]≜\left[\begin{array}{ccc} 0 & -p_{3} & p_{2}\\ p_{3} & 0 & -p_{1}\\ -p_{2} & p_{1} & 0 \end{array}\right].$$

### 4.2. The Proposed Dual Foot-Mounted IMU Algorithm

The dual foot-mounted IMU errors state vector is defined by combining the left and right foot error states:(21)$$X_{e}={\left[\begin{array}{cc} x_{e,L}^{T} & x_{e,R}^{T} \end{array}\right]}^{T}\in R^{18}.$$

The dynamic equation is expressed as follows:(22)$${\dot{X}}_{e}=Fx+W,$$
where $F=\left[\begin{array}{cc} A_{L} & 0_{9}\\ 0_{9} & A_{R} \end{array}\right]$ and $W=\left[\begin{array}{c} w_{L}\\ w_{R} \end{array}\right]$.

Let τ be the sampling period of the inertial sensor output. The system equation is discretized as follows:(23)$$X_{e,k+1}=\Phi_{k}X_{e,k}+W_{k},$$
where $\Phi_{k}=e^{F\tau }\approx I_{18}+F\tau +0.5{\left(F\tau \right)}^{2}$ and $Q_{k}=E\left\{W_{k}W_{k}^{T}\right\}$.

In the proposed system, the model and observation equations are linear except the quaternion model, which is approximately linearized. The divergence can be the direct consequence of modeling errors. In the previous initial heading estimation section, although the initial heading of the INS system is unknown and has large uncertainty, it does not affect our proposed system. Three measurement updates are used in this Kalman filter: the zero-velocity update, the LiDAR-based stance foot position update, and the straight-walking-based heading and foot position update.

#### 4.2.1. Zero-Velocity Update

If the left/right foot is in the stance phase, the corresponding measurement equations are given by:(24)$$\begin{array}{cc} z_{v,L} & =0_{3\times 1}-{\hat{v}}_{L,k}=H_{v,L}X_{e,k}+n_{v,L},\\ z_{v,R} & =0_{3\times 1}-{\hat{v}}_{R,k}=H_{v,R}X_{e,k}+n_{v,R}, \end{array}$$
where $H_{v,L}=\left[\begin{array}{cccccc} 0_{3} & 0_{3} & I_{3} & 0_{3} & 0_{3} & 0_{3} \end{array}\right]$, $H_{v,R}=\left[\begin{array}{cccccc} 0_{3} & 0_{3} & 0_{3} & 0_{3} & 0_{3} & I_{3} \end{array}\right]$, $n_{v,L}$ and $n_{v,R}$ are the white Gaussian measurement noises, and $R_{v}=E\left\{n_{v,j}n_{v,j}^{T}\right\}$.

#### 4.2.2. Lidar-Based Stance Foot Position Update

This update is used in the stance phase of each foot if the stance leg position estimation of LiDAR $p_{i,j}$ (i=1,2 and j=L,R) is available. Here, the LiDAR-based stance leg positions are already transformed into the LiDAR 1 coordinate system, so that we ignore the right superscript $L_{1}$ in $p_{i,j}$.

As we mention in the Section 3.3, we assume that the human leg is vertically straight during the mid-stance phase; therefore, we can transform the LiDAR-based stance leg center point position to the x−y coordinate of stance foot position $f_{i,j}$ using the following equation:(25)$$\left[\begin{array}{c} f_{i,j}\\ {}* \end{array}\right]=\left[\begin{array}{c} p_{i,j}\\ 0 \end{array}\right]+{\left(C\left({\hat{q}}_{j}\right)\right)}^{T}r_{K_{j}}^{I_{j}},$$
where ${\hat{q}}_{j}$ is the currently estimated quaternion of IMU *j*, and $r_{K_{j}}^{I_{j}}$ is the pre-calibrated coordinate of ankle joint in IMU *j* coordinate, as demonstrated in Section 3.3. Now, we can update the x−y coordinate of the stance foot using the following measurement equations:(26)$$\begin{array}{cc} z_{lidar,L} & =f_{i,L}-\left[\begin{array}{c} {\hat{r}}_{L,x}\\ {\hat{r}}_{L,y} \end{array}\right]=H_{r,L}X_{e,k}+n_{lidar,L},\\ z_{lidar,R} & =f_{i,R}-\left[\begin{array}{c} {\hat{r}}_{R,x}\\ {\hat{r}}_{R,y} \end{array}\right]=H_{r,R}X_{e,k}+n_{lidar,R}, \end{array}$$
where $H_{r,L}=\left[\begin{array}{cccccc} 0_{2\times 3} & \left[\begin{array}{ccc} 1 & 0 & 0\\ 0 & 1 & 0 \end{array}\right] & 0_{2\times 3} & 0_{2\times 3} & 0_{2\times 3} & 0_{2\times 3} \end{array}\right]$,

$H_{r,R}=\left[\begin{array}{cccccc} 0_{2\times 3} & 0_{2\times 3} & 0_{2\times 3} & 0_{2\times 3} & \left[\begin{array}{ccc} 1 & 0 & 0\\ 0 & 1 & 0 \end{array}\right] & 0_{2\times 3} \end{array}\right]$, $n_{lidar,L}$, and $n_{lidar,R}$ are the white Gaussian measurement noises with covariance $R_{lidar}$.

#### 4.2.3. Heading and Position Update during Straight Walking

Instead of using the pre-defined building heading, we update the current walking stride heading using the previous stride headings if straight walking is detected. The new position of current stance foot is then calculated based on this heading. Then, it is used as the measurement update in the Kalman filter. Since we use the heading of previous strides, the heading errors are still accumulated until the measurement data of LiDAR 2 is available. These errors are compensated later in the smoother algorithm.

Figure 7 shows an example of the proposed algorithm for current stance left foot. Let ${\hat{r}}_{L,s}$ denote the current estimated left foot position, where *s* is the discrete index of the current stance phase. The current walking stride on the right side and the adjacent walking stride on the left side are used. Their stance foot positions are represented by ($r_{R,s}$, $r_{R,s-1}$), and ($r_{L,s-1}$, $r_{L,s-2}$), respectively. The straight detection and heading correction procedures are as follows:Condition 1: The current step length is calculated:
$$∥{\hat{S}}_{LR}∥=∥{\hat{r}}_{L,s}-r_{R,s}∥.$$Only the walking step with $∥{\hat{S}}_{LR}∥>\delta_{s}$ is considered. Here, the threshold is empirically chosen to check whether a normal walking step has indeed been taken.Condition 2: If the current walking step is normal, we compute the angles between the current stride vector ${\hat{S}}_{L,c}={\hat{r}}_{L,s}-r_{L,s-1}$ and the previous left stride vector $S_{L}=r_{L,s-1}-r_{L,s-2}$ and the right stride vector $S_{R}=r_{R,s}-r_{R,s-1}$:
$$\alpha_{1}=∠\left({\hat{S}}_{L,c},S_{L}\right),\;\alpha_{2}=∠\left({\hat{S}}_{L,c},S_{R}\right).$$If they are smaller than a threshold $\delta_{\alpha }$, straight walking is detected. Here, we use the same threshold values used in [22], i.e., $\delta_{s}=0.5$ m and $\delta_{\alpha }=8^{o}$.The current stride vector is set as the average value of previous left and right stride vectors ${\hat{S}}_{L,c,new}=\mathrm{mean}\left\{s_{L},s_{R}\right\}$. Then, we normalize the ${\hat{S}}_{L,c,new}$ and update the new position of the current left foot as follows:
$${\hat{r}}_{L,s,new}=r_{L,s-1}+∥{\hat{S}}_{L,c}∥\frac{{\hat{S}}_{L,c,new}}{∥{\hat{S}}_{L,c,new}∥}.$$Similarly, if the right straight walking step is detected, the new position of the current right foot ${\hat{r}}_{R,s,new}$ can be estimated.

Using the new current left/right foot positions, the measurement equations are given by:(27)$$\begin{array}{cc} z_{p,L} & ={\hat{r}}_{L,s,new}-\left[\begin{array}{c} {\hat{r}}_{L,x}\\ {\hat{r}}_{L,y} \end{array}\right]=H_{r,L}X_{e,k}+n_{p,L},\\ z_{p,R} & ={\hat{r}}_{R,s,new}-\left[\begin{array}{c} {\hat{r}}_{R,x}\\ {\hat{r}}_{R,y} \end{array}\right]=H_{r,R}X_{e,k}+n_{p,R}, \end{array}$$
where $H_{r,L}$ and $H_{r,R}$ are given in (Equation 26), and $n_{p,L}$ and $n_{p,R}$ are the white Gaussian measurement noises with covariance $R_{p}$. Note that we only update the x−y positions in this part.

In the Kalman filter, $X_{e,k}$ is estimated and used to compensate the errors in $\hat{q}$, $\hat{r}$, and $\hat{v}$ using (Equation 18). After updating, the errors are set to zero.

### 4.3. Smoother Algorithm

Since we use the estimated previous heading to update the stride heading in straight walking, the heading errors are still accumulated. To archive our proposed correction performance, the errors of the Kalman filter estimated values (denoted by ${\hat{q}}_{KF,k}$, ${\hat{v}}_{KF,k}$, and ${\hat{v}}_{KF,k}$) are compensated using the smoother algorithm [32]. For that, we define the estimation error-state vector as follows:(28)$$X_{SM,k}=\left[\begin{array}{c} {\bar{q}}_{L,k}\\ {\bar{r}}_{L,k}\\ {\bar{v}}_{L,k}\\ {\bar{q}}_{R,k}\\ {\bar{r}}_{R,k}\\ {\bar{v}}_{R,k} \end{array}\right]=\left[\begin{array}{c} \left[\begin{array}{cc} 0_{3\times 1} & I_{3} \end{array}\right]{\hat{q}}_{KF,L,k}^{*}⊗q_{L,k}\\ r_{L,k}-{\hat{r}}_{KF,L,k}\\ v_{L,k}-{\hat{v}}_{KF,L,k}\\ \left[\begin{array}{cc} 0_{3\times 1} & I_{3} \end{array}\right]{\hat{q}}_{KF,R,k}^{*}⊗q_{R,k}\\ r_{R,k}-{\hat{r}}_{KF,R,k}\\ v_{R,k}-{\hat{v}}_{KF,R,k} \end{array}\right].$$

The system equation for smoother estimation values is given by:(29)$$\zeta_{k}+X_{SM,k+1}=\Phi_{k}X_{SM,k}+W_{k},$$
where $\Phi_{k}$ is defined in (Equation 23) and
(30)$$\zeta_{k}=\left[\begin{array}{c} {\tilde{q}}_{SM,L,k}\\ {\hat{r}}_{KF,L,k+1}-f_{2,k}\\ {\hat{v}}_{KF,L,k+1}-f_{3,k}\\ {\tilde{q}}_{SM,R,k}\\ {\hat{r}}_{KF,R,k+1}-f_{5,k}\\ {\hat{v}}_{KF,R,k+1}-f_{6,k} \end{array}\right],$$
(31)$$\left[\begin{array}{c} f_{1,k}\\ f_{2,k}\\ f_{3,k}\\ f_{4,k}\\ f_{5,k}\\ f_{6,k} \end{array}\right]=f_{k}\left(\left[\begin{array}{c} {\hat{q}}_{KF,L,k}\\ {\hat{r}}_{KF,L,k}\\ {\hat{v}}_{KF,L,k}\\ {\hat{q}}_{KF,R,k}\\ {\hat{r}}_{KF,R,k}\\ {\hat{v}}_{KF,R,k} \end{array}\right],\left[\begin{array}{c} v_{g,L,k}\\ v_{a,L,k}\\ v_{g,R,k}\\ v_{a,R,k} \end{array}\right]\right),$$
$${\tilde{q}}_{SM,L,k}=\left[\begin{array}{cc} 0_{3\times 1} & I_{3} \end{array}\right]f_{1,k}^{*}⊗{\hat{q}}_{KF,L,k+1},$$
$${\tilde{q}}_{SM,R,k}=\left[\begin{array}{cc} 0_{3\times 1} & I_{3} \end{array}\right]f_{4,k}^{*}⊗{\hat{q}}_{KF,R,k+1}.$$

Function $f_{k}$ in (Equation 31) is the numerical integration of quaternion, position, and velocity from kτ to (k+1)τ. Using (Equation 28), (Equation 30) and (Equation 31), we can expand the left side of (Equation 29) as the errors of the numerical integration of the Kalman filter estimated values. Thus, (Equation 29) represents how the estimation errors evolve after the integration of (Equation 31).

The smoothing problem can be formulated as a quadratic optimization problem using the method in [32]. Let the optimization variable $\tilde{X}$ be defined by:(32)$$\tilde{X}=\left[\begin{array}{c} X_{SM,1}\\ X_{SM,2}\\ \vdots \\ X_{SM,N} \end{array}\right]\in R^{18N\times 1}.$$

We can estimate $\tilde{X}$ by minimizing the cost function:(33)$$\begin{array}{cc} J(\tilde{X})\; & =\frac{1}{2}\sum_{k=1}^{N-1}{\left(\zeta_{k}+X_{SM,k+1}-\Phi_{k}X_{SM,k}\right)}^{T}Q_{k}^{-1}\left(\zeta_{k}+X_{SM,k+1}-\Phi_{k}X_{SM,k}\right)\\ \; & +\frac{1}{2}{\left(X_{SM,1}-X_{init}\right)}^{T}P_{init}^{-1}\left(X_{SM,1}-X_{init}\right)\\ \; & +\mathrm{constraint}\;\mathrm{terms}\;\mathrm{in}\;(36), \end{array}$$
where $P_{init}=\left[\begin{array}{cc} P_{L,init} & 0_{9}\\ 0_{9} & P_{R,init} \end{array}\right]$ in which $P_{j,init}=\left[\begin{array}{ccc} P_{q_{j,0}} & 0_{3} & 0_{3}\\ 0_{3} & P_{r_{j,0}} & 0_{3}\\ 0_{3} & 0_{3} & P_{v_{j,0}} \end{array}\right],\;j=L,R$. The initial attitude error covariance $P_{q_{j,0}}$ is given by the algorithm in [33]. The initial position and velocity error covariance $P_{r_{j,0}}$, $P_{v_{j,0}}$ can be considered as design parameters.

In this smoother algorithm, we use three measurements: the zero-velocity update, the LiDAR-based stance foot positions update, and the step width constraint update. The first two measurements are explained in the previous section of the Kalman filter. The third measurement is proposed using an assumption that the width of walking step is almost constant during straight walking.

As an example, during a current stance phase of left foot, the step width is estimated as follows:The walking step vector is computed: $S_{LR}={\hat{r}}_{KF,L,k}-{\hat{r}}_{KF,R,k}$;The orthogonal unit vector $n_{L}$ of the current walking direction vector ${\hat{S}}_{L,c,new}$ that points toward opposite foot side is computed, so that $n_{L}^{T}S_{LR}>0$;The step width *d* is now calculated as follows:
$$d=n_{L}^{T}S_{LR}=n_{L}^{T}\left({\hat{r}}_{KF,L,k}+{\bar{r}}_{L,k}-{\hat{r}}_{KF,R,k}-{\bar{r}}_{R,k}\right).$$

The step width measurement equation for the left stance foot is given as follows:(34)$$z_{d,L,k}=H_{d,L}X_{SM,k}+n_{d,L},$$
where $z_{d,L,k}=d-n_{L}^{T}\left({\hat{r}}_{KF,L,k}-{\hat{r}}_{KF,R,k}\right)$ and



$H_{d,L}=\left[\begin{array}{cccccc} 0_{1\times 3} & \left[\begin{array}{cc} n_{L}^{T} & 0 \end{array}\right] & 0_{1\times 3} & 0_{1\times 3} & \left[\begin{array}{cc} -n_{L}^{T} & 0 \end{array}\right] & 0_{1\times 3} \end{array}\right].$



Similarly, the step width measurement equation for the right stance foot is given as follows:(35)$$z_{d,R,k}=H_{d,R}X_{SM,k}+n_{d,R},$$
where $z_{d,R,k}=d-n_{R}^{T}\left({\hat{r}}_{KF,R,k}-{\hat{r}}_{KF,L,k}\right)$ and

$H_{d,R}=\left[\begin{array}{cccccc} 0_{1\times 3} & \left[\begin{array}{cc} -n_{R}^{T} & 0 \end{array}\right] & 0_{1\times 3} & 0_{1\times 3} & \left[\begin{array}{cc} n_{R}^{T} & 0 \end{array}\right] & 0_{1\times 3} \end{array}\right]$. In (Equation 34) and (Equation 35), $n_{d,L}$ and $n_{d,R}$ are the white Gaussian measurement noises with covariance $R_{d}$.

Consequently, the “constraint terms” in (Equation 33) are detected as follows:(36)$$\begin{array}{cc} \mathrm{constraint}\;\mathrm{terms} & =\frac{1}{2}\sum_{i=L,R}\\ \; & (\sum_{k\in Z_{v,i}}{\left(z_{v,i,k}-H_{v,i,k}X_{SM,i,k}\right)}^{T}R_{v}^{-1}\left(z_{v,i,k}-H_{v,i,k}X_{SM,i,k}\right)+\\ \; & \sum_{k\in Z_{r,i}}{\left(z_{r,i,k}-H_{r,i,k}X_{SM,i,k}\right)}^{T}R_{lidar}^{-1}\left(z_{r,i,k}-H_{r,i,k}X_{SM,i,k}\right)+\\ \; & \sum_{k\in Z_{d,i}}{\left(z_{d,i,k}-H_{d,i,k}X_{SM,i,k}\right)}^{T}R_{d}^{-1}\left(z_{d,i,k}-H_{d,i,k}X_{SM,i,k}\right)), \end{array}$$
where $Z_{v,i}$, $Z_{r,i}$, and $Z_{d,i}$ denote the sets of all discrete time indices belonging to zero-velocity intervals, LiDAR-based stance foot position measurement intervals, and step width measurement intervals, respectively. The cost function (Equation 33) can be rewritten as a quadratic function of $\tilde{X}$ as follows:(37)$$J\left(\tilde{X}\right)=\frac{1}{2}{\tilde{X}}^{T}M_{1}\tilde{X}+M_{2}\tilde{X}+M_{3},$$
where $M_{3}$ is irrelevant in the optimization, and $M_{1}$ and $M_{2}$ can be easily computed using (Equation 33) and (Equation 36). The details can be found in [32]. This is the main advantage of the quadratic-based optimization smoother algorithm, where the measurement constraints can be easily included in the optimization problem. The minimizing solution of (Equation 37) can be computed by solving the following equation:(38)$$M_{1}{\tilde{X}}^{*}+M_{2}=0,$$
where ${\tilde{X}}^{*}$ is the minimizing solution. Once ${\tilde{X}}^{*}$ is computed, the smoother estimation values ${\hat{q}}_{SM,L,k}$, ${\hat{r}}_{SM,L,k}$, ${\hat{v}}_{SM,L,k}$, ${\hat{q}}_{SM,R,k}$, ${\hat{r}}_{SM,R,k}$, and ${\hat{v}}_{SM,R,k}$ can be found using (Equation 28).

## 5. Experimental Results

### 5.1. Hardware Description

A Xsens MTi-1 IMU module with a micro SD card is installed in each foot. The sampling rate of two IMUs is 100 Hz. Two LiDARs (RPLIDAR A3) have a scan rate of 10 Hz. The total cost of the system is around USD 1500 (USD 150 for an IMU and USD 600 for a LiDAR). The sensors specifications are shown in Table 1. An OptiTrack optical motion capture system, which is composed of six Flex 13 cameras with a resolution of 1280×1024 at 120 Hz, is used to provide the ground truth of the walking step length estimation.

The parameters used in this paper are given in Table 2.

### 5.2. Stride Length Estimation Evaluation

To evaluate the stride length estimation, we perform a straight walking experiment with the optical camera system. Two male healthy volunteers, which have no problems with gait, balance, and coordination, are recruited. Their information is given in Table 3. They are asked to walk along the corridor with the travel distances of 20 m and 50 m. Each person repeats the walk 10 and 5 times in 20 m and 50 m experiments, respectively.

The configuration of the experiment is given in Figure 8. The optical system cannot cover all the walking path; thus, it is placed so that a walking path outside two LiDARs range can be captured. The walking path outside the LiDARs range has the least measurement update; therefore, the walking stride length errors may be largest. With this setup, our OptiTrack system can capture a walking range of 8–13 m measured from LiDAR 2 origin. By comparing the walking stride lengths captured in this path, the performance of our proposed system is evaluated. The LiDAR 2 and OptiTrack system positions are fixed during both experiments. The LiDAR 1 position is changed with the walking distance.

The subject may have difficulties in following the straight path during walking. We model a simple case of abnormal gait by removing the availability of straight walking in 5 s (20 m walking) and 10 s (50 m walking). The straight-walking-based foot position update is therefore not applied in this interval. The step width constraint is still used with larger uncertainty. The modeled measurement update is then fed to the proposed method. An example of measurement availability of two feet is given in Figure 9. The top plot is the normal availability, while the bottom one is the modeled availability of measurement update. The non-zero values represent the presence of measurements. We can see the periodicity of zero-velocity measurements during walking. The LiDAR 1-based stance foot position measurements are available for the first three and four stance phases of the left and right foot, respectively. LiDAR 2 provides the stance foot position update for the last four and three stance phases of the left and right foot, respectively. In the stance phases where the LiDAR data are not available, the straight-walking-based heading and position update (in the Kalman filter) or the step width constraint update (in the smoother algorithm) is integrated. As in the bottom plot, the straight-walking-based measurement update is not available during the interval from 15 to 25 s.

The two estimated feet trajectories of all 20 m and 50 m walking experiments using the normal and modeled update are given in Figure 10 and Figure 11. Figure 12 shows an example of a single 50 m walking estimation result for clearer vision. The LiDAR-based stance foot position estimation is denoted by a circle. We can see that the Kalman filter with the initial heading estimated from Section 3.3 and the foot position update during straight walking reduces the heading errors significantly. However, the heading errors can still be observed as the estimated trajectories do not match with the foot position update from LiDAR 2. This error is compensated using the smoother algorithm, where the step width constraint is used instead of using the heading and foot position update. If the heading and foot position update is used, the Kalman-filer-based heading estimation will be maintained in the smoother algorithm. Therefore, the heading errors are not compensated. Based on the heading estimation from the Kalman filter, the advantage of dual foot-mounted IMU is expressed in the step width constraint. From the all estimated walking trajectories results, we can see a small bulge in the middle of the walk. The further walking distance is, the bigger bulge is. This is because there is no actual heading observation outside the LiDARs’ range. However, with the measurement update from LiDAR 2, the heading estimation errors can be corrected. In 20 m and 50 m walking estimation, the results using the normal and modeled update are almost similar. This shows the robustness of the proposed method.

The walking stride length estimation is then evaluated. For each walking, the optical OptiTrack system can capture three to four walking strides for each foot. The total counts of captured walking strides are 98 and 101 for the left and right foot, respectively. We compare the estimated walking stride length from the proposed smoother algorithm ($SL_{IMU}$) with the optical estimation ($SL_{optical}$) by computing the errors:$$e=SL_{optical}-SL_{IMU}.$$

Figure 13 shows the histogram of the stride length estimation errors. Table 4 shows the mean estimation of the stride length and variance of each subject with a normal measurement update. The estimation errors compared to the optical system estimation are also given. For all subjects, the mean and root mean square errors (RMSE) of stride length estimation are less than 3 cm, while the maximum errors are less than 8 cm. The performances of the stride length estimation of the normal measurement update and the modeled measurement update are almost identical.

### 5.3. Evaluation of Rectangular Path Walking Estimation

To verify the effectiveness of the proposed method in heading correction, a rectangular path walking experiment is carried out. In this experiment, a person starts from the starting point, walks counter-clockwise twice along a rectangular path, and stops at the ending point as in Figure 14. The total of walking distance is around 124 m, with six turning phases. The LiDARs are placed to be able to capture the human leg data during turning phases.

Figure 15 shows the left and right foot estimated trajectories results of the proposed algorithm. We plot the corridor wall scan data as a reference of the walking direction as well as the results of calibrating the two LiDARs. The shading circle area represents the human leg tracking range of each LiDAR. As we can see, the proposed method can correct the heading during turning phases using the LiDAR measurements. There is no intersection between two feet trajectories. For comparison, we also plot the result of the dual foot-mounted IMU algorithm, which is proposed in [22], as in Figure 16. The algorithm using only dual foot-mounted IMU with a foot position update during straight walking based on the maximum range of the one-side foot stride length and the both-sides feet step length. Since there is no measurement of the heading, the errors can be seen clearly after the second round.

## 6. Conclusions

This paper proposes a human gait tracking system using a dual foot-mounted IMU and multiple 2D LiDARs. Two LiDARs are placed far apart: LiDAR 1 is at the starting point and LiDAR 2 is at the destination in straight walking or at the turning point in rectangular path walking. A calibration algorithm is proposed to estimate the rotation angle and translation vector of two LiDARs. The calibration uses a cylinder placed at four positions which are known their geometrical information. An iterative algorithm is then used to estimate the calibration parameters with the including of LiDAR measurement errors and configuration errors. Then, we can use the LiDARs as anchors to correct the inertial sensor system estimation. The LiDAR-based stance leg position estimations are transformed to stance foot position, and then combined as the measurement update of the Kalman filter and the smoother algorithm.

Since the LiDAR can provide the environment information, the LiDAR 1 coordinate system is chosen as the global coordinate system. The initial heading and position of each IMU can be estimated using IMUs data and the leg scan data from LiDAR 1. For further improvement, we update the heading and foot position if straight walking is detected. This is used as a measurement update in the Kalman filter. In the smoother algorithm, the straight-walking-based foot position update is replaced by a relative step width constraint update. We assume that in normal walking, the width of walking step is almost constant. We verify the stride length estimation of the proposed system with straight walking of 20 m and 50 m. We also model the abnormal walking by assuming that the subject cannot follow the straight path during a time interval. The measurement update of straight walking is therefore not available during this interval. The root mean square errors of the walking stride length estimation of both normal and modeled update are less than 3 cm compared with an optical tracking system. Through the rectangular path walking experimental results, we can see the LiDAR measurement can be used to correct the heading effectively. The proposed system has the potential for use in many fields, such as human gait monitoring and human gait navigation systems.

## Figures and Tables

**Figure 1 sensors-22-06368-f001:**
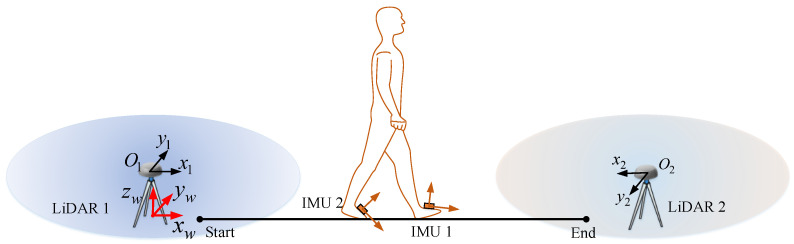
System overview: a dual foot-mounted IMU and multiple 2D LiDARs.

**Figure 2 sensors-22-06368-f002:**
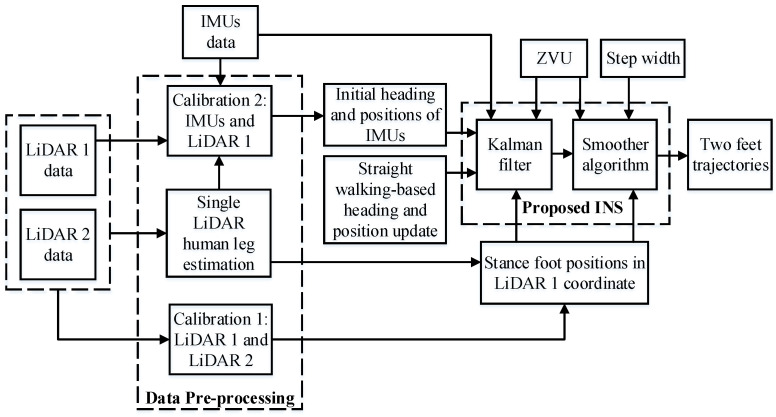
Block diagram of proposed method.

**Figure 3 sensors-22-06368-f003:**
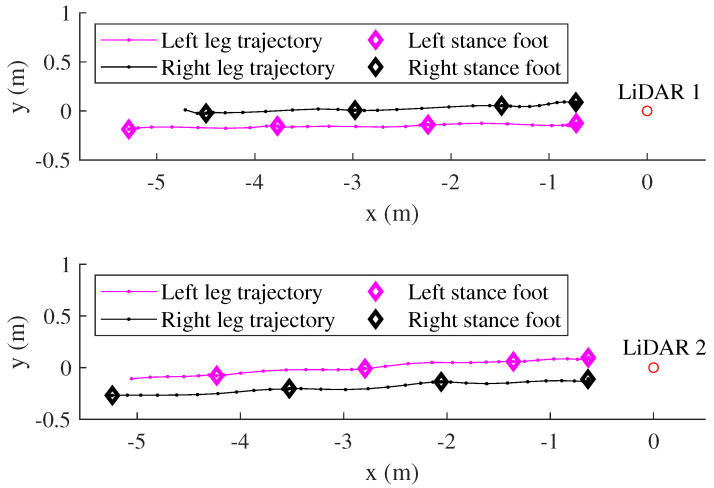
Estimation of the human leg trajectories from each LiDAR.

**Figure 4 sensors-22-06368-f004:**
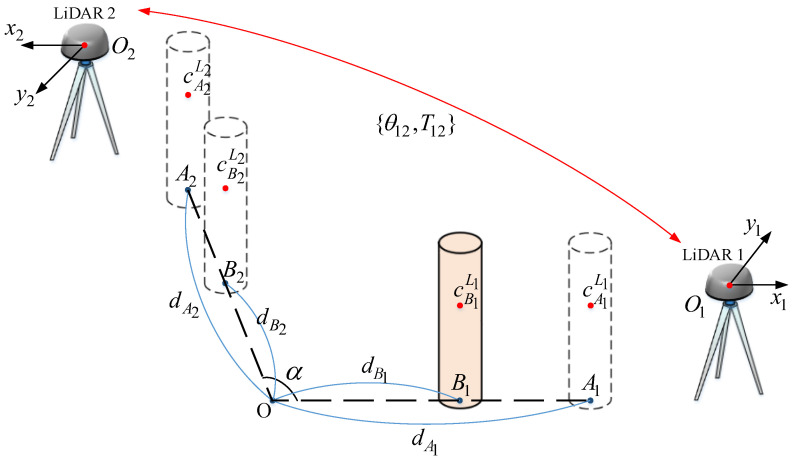
The configuration of LiDARs calibration.

**Figure 5 sensors-22-06368-f005:**
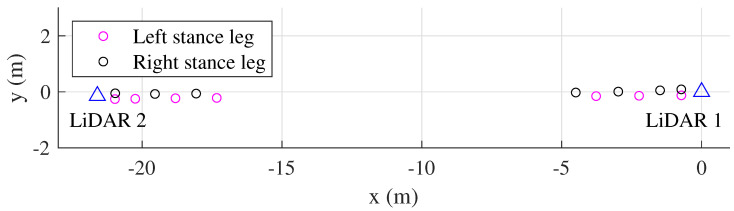
Two LiDARs calibration results in 20 m straight walking. The stance leg positions from LiDAR 2 are transformed into the LiDAR 1 coordinate system.

**Figure 6 sensors-22-06368-f006:**
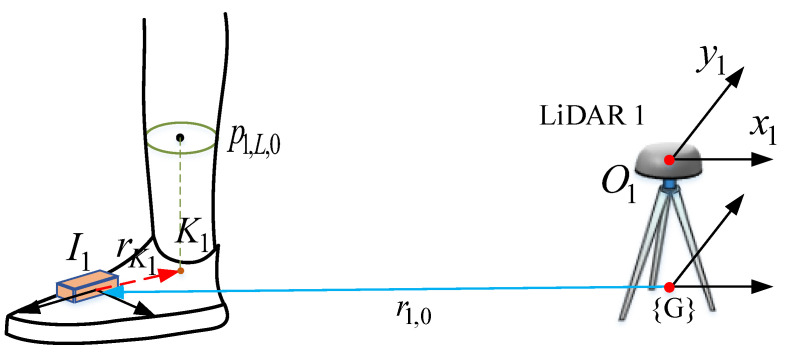
Initial position of IMU in LiDAR 1 coordinate estimation.

**Figure 7 sensors-22-06368-f007:**
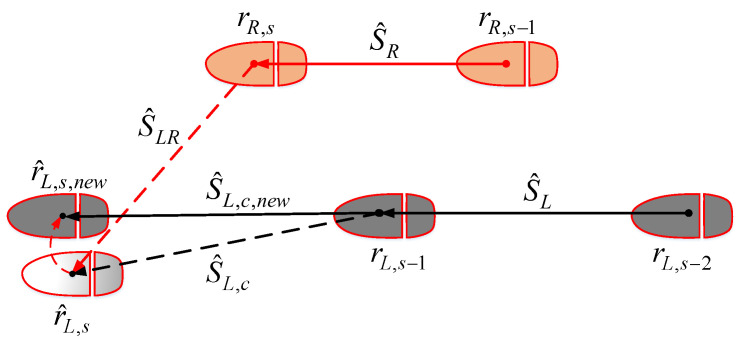
Straight detection for current left foot.

**Figure 8 sensors-22-06368-f008:**
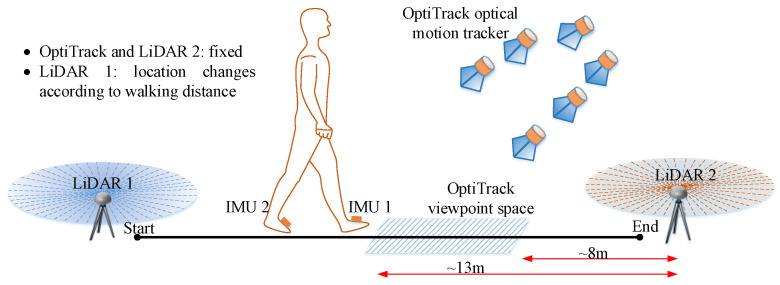
The configuration of the 20 m and 50 m walking distance experiment.

**Figure 9 sensors-22-06368-f009:**
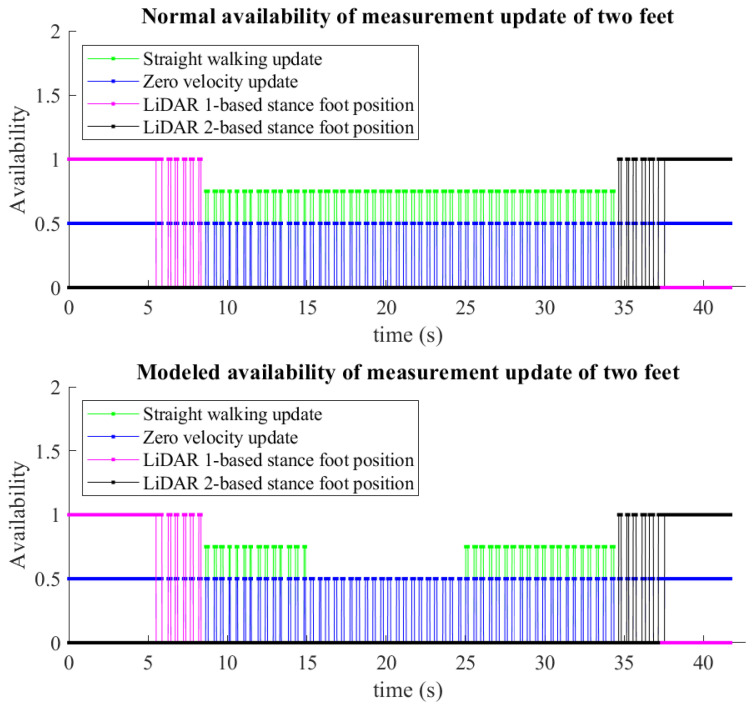
An example of measurement availability of two feet in 50 m walking. The straight-walking-based measurement updates are available for all walking steps outside the LiDAR range in the normal update (**top** plot) and are removed by 10 s in the modeled update (**bottom** plot).

**Figure 10 sensors-22-06368-f010:**
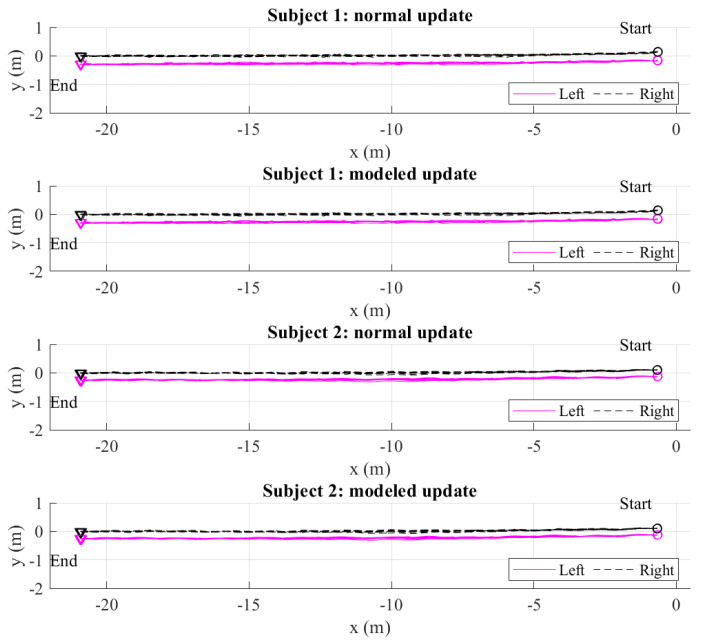
The total estimated 20 m walking trajectories of two subjects from the smoother algorithm in the normal and modeled update.

**Figure 11 sensors-22-06368-f011:**
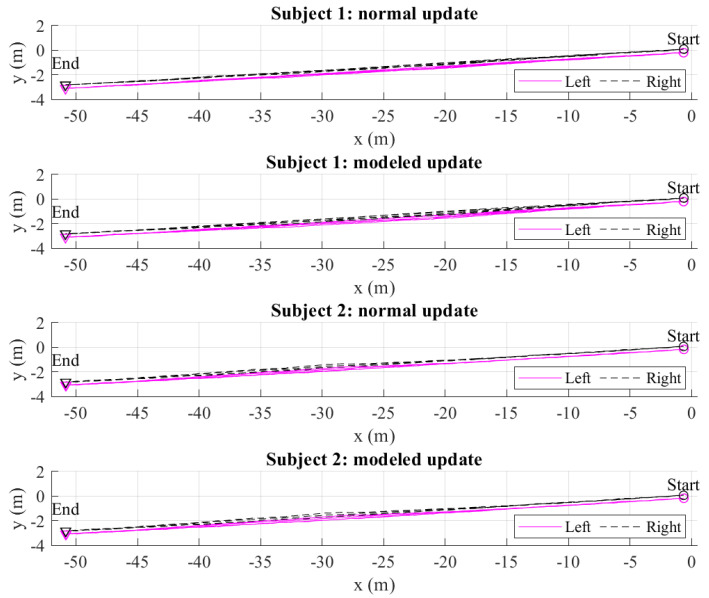
The total estimated 50 m walking trajectories of two subjects from the smoother algorithm in the normal and modeled update.

**Figure 12 sensors-22-06368-f012:**
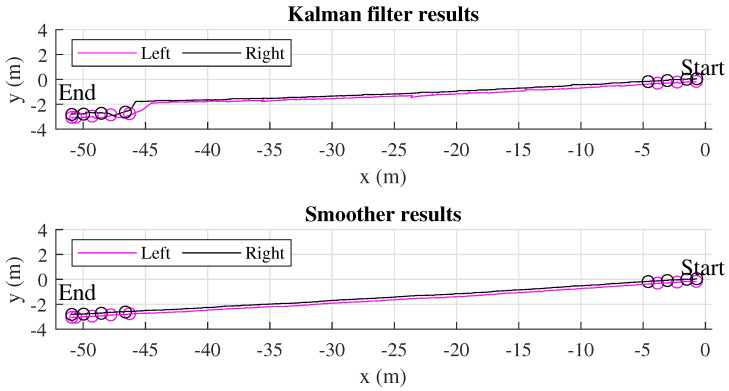
An example of an estimation of 50 m walking trajectories from the Kalman filter and the smoother algorithm. The circles represent the estimated LiDAR-based foot positions in the stance phase.

**Figure 13 sensors-22-06368-f013:**
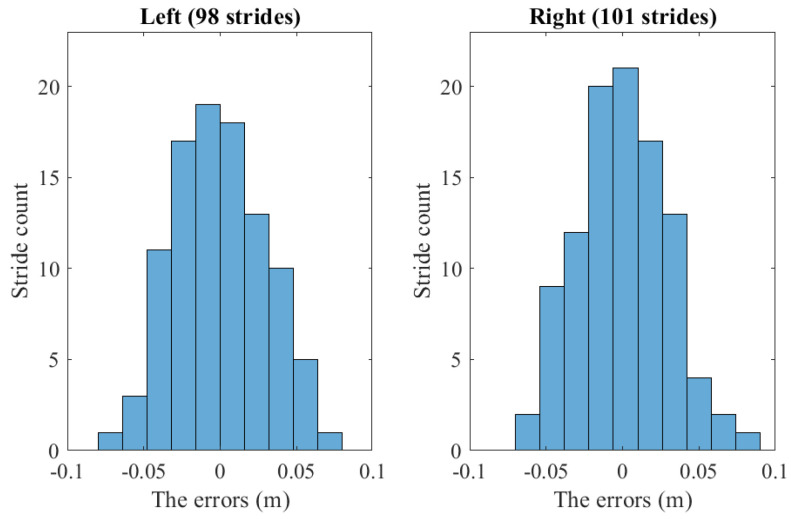
The histogram of all walking stride length estimation errors of the proposed method.

**Figure 14 sensors-22-06368-f014:**
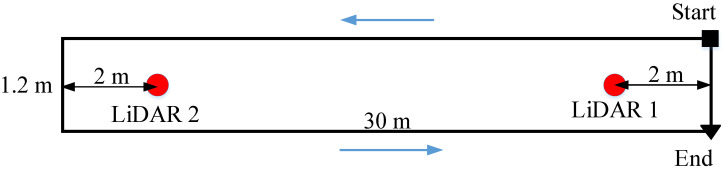
The configuration of rectangular path walk experiment.

**Figure 15 sensors-22-06368-f015:**
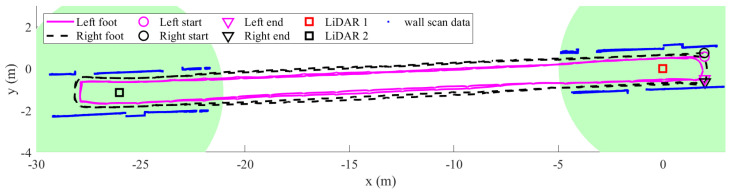
The estimated trajectory of proposed method in the rectangular path walking experiment. The shading circle area indicates the human leg tracking range of each LiDAR.

**Figure 16 sensors-22-06368-f016:**
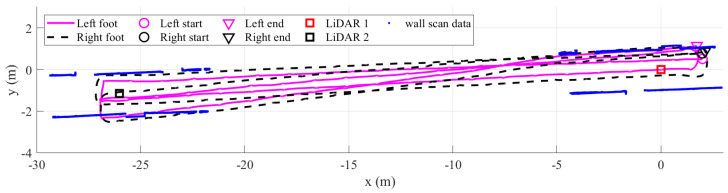
The estimated trajectory inthe rectangular path walking experiment, in which the LiDAR data are not used.

**Table 1 sensors-22-06368-t001:** Sensor specifications.

Specification	Xsens MTi-1	
	Accelerometer	Gyroscope
Full range	±16g	$\pm 2000^{\circ }/s$
Noise density	120 μg/√Hz	$0.007^{\circ }/s/\surd \mathrm{Hz}$
Bandwidth	324 Hz (z: 262 Hz)	255 Hz
Sampling rate	100 Hz	100 Hz
**Specification**	**RPLIDAR A3**	
Distance range	10–25 m	
Minimum operating ranging	0.2 m	
Angular resolution	$0.225^{\circ }$	
Accuracy	1% of range ≤3 m	
	2% of range 3–5 m	
	2.5% of range 5–25 m	
Scan rate	10 Hz	

**Table 2 sensors-22-06368-t002:** Parameters used in this paper.

Parameters	Related Equations
$R_{a}=0.01I_{3}$, $R_{g}=0.001I_{3}$	(Equation 2)
$R_{r_{c}}=0.01$, $R_{\beta }=0.01$, $R_{\alpha }=0.01$	(Equation 9)
γ=0.001	(Equation 12)
$B_{g}=1.2$, $B_{a}=1.5$, $N_{g}=16$, $N_{a}=16$	(Equation 17)
$P_{r_{j,0}}=0.000001I_{3}$, $P_{v_{j,0}}=0.000001I_{3}$	(Equation 33)
$R_{d}=0.01$	(Equation 9) and (Equation 36)
$R_{v}=0.01I_{3}$	(Equation 24)
$R_{lidar}=0.01I_{2}$	(Equation 26)
$R_{p}=0.01I_{2}$	(Equation 27)

**Table 3 sensors-22-06368-t003:** Subjects information.

Subject	Age	Height (cm)	Weight (kg)
1	33	168	63
2	30	160	52

**Table 4 sensors-22-06368-t004:** The stride length estimation results compared with the optical OptiTrack system.

Subject	Mean Stride Length (m)	Estimation Errors			
		Leg (Strides)	Max (cm)	Mean (cm)	RMSE (cm)
1	1.503 ± 0.054	Left (45)	7.5	2.3	2.8
	1.489 ± 0.047	Right (56)	6.2	2.1	2.5
2	1.416 ± 0.032	Left (53)	7.6	2.7	3.2
	1.377 ± 0.056	Right (45)	6.4	2.5	3.1
All	Normal update	Left (98)	7.6	2.5	3.0
		Right(101)	6.4	2.3	2.8
	Modeled update	Left (98)	7.6	2.5	3.1
		Right(101)	6.4	2.3	2.8

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
