# Peer review of "A Human Gait Tracking System Using Dual Foot-Mounted IMU and Multiple 2D LiDARs"

_sensors, 2022, doi:10.3390/s22176368_

Round 1

Reviewer 1 Report

My primary concern is in proofreading. Portions of the manuscript are good, but others have issues with preposition and article use. Finding a proofreader with a strong background in English writing and a thorough review of the article will help overall readability. 

This design is innovative, and has the potential to have a significant impact in medical testing in clinical, occupational, and military settings. 

I think that an n of 2 "healthy volunteers" is less rigorous that I would like to see. Adding information about height, weight, age, and definition of "healthy" would be useful. Comparison of the data for the 2 healthy subjects would also be appreciated. 

Although the current study appears to be focused on feasibility and optimization, some discussion about applications of this tool for diagnostic purposes would be appreciated. I believe that you have the data that you need to model injury/disability/disease. I wonder how this system would analyze a subject recovering from a stroke, spinal cord injury, vestibular lesion, or learning to walk with a prosthetic limb. A specific question is what would happen if a patient was having difficulty lifting one leg, or if the two legs did not follow similar paths during walking. This might be a "future direction", or a study that you are already performing, but providing a model that accounts for abnormal gait would be a significant bonus in this paper (coming from a basic scientist interested in evaluation of motor performance after injury).  

Reviewer 2 Report

The paper provides a human gait tracking system using a dual foot-mounted IMU and multiple 2D LiDARs to overcome the known disadvantages of each single sensor system. The paper is interesting and generally well written. However, some issues need to be solved:

1.       The selection of the two thresholds from line 308 needs to be better explained. Why are they relevant for this problem? What happens if, for example a 0.4m threshold (delta_s) is chosen instead of 0.5?

2.       It is known that Kalman filter has two limitations: a) It assumes that both the system and observation models equations are both linear; and b) It assumes that the state belief is Gaussian distributed. Please include a short discussion about these two limitations and how they may affect the proposed solution.

3.    Please add the required Back matter (see “Instructions for Authors”) just before References list (Author Contributions, Conflicts of Interest, etc.).

Round 2

Reviewer 1 Report

Thank you for incorporating my suggestions. I look forward to seeing future studies of abnormal gait using this design. I have no additional suggestions.

Author Response

Thank you for your kind suggestions.

Reviewer 2 Report

The authors have successfully solved my comments and concerns.

Author Response

Thank you for your kind suggestions.